# BEYOND FINITE DATA: TOWARDS DATA-FREE OUT-OF-DISTRIBUTION GENERALIZATION VIA EXTRAPOLATION

## ABSTRACT

Out-of-distribution (OOD) generalization is a favorable yet challenging property for deep neural networks. The core challenges lie in the limited availability of source domains that help models learn an invariant representation from the spurious features. Various domain augmentation have been proposed but largely rely on interpolating existing domains and frequently face difficulties in creating truly "novel" domains. Humans, on the other hand, is capable of extrapolating novel domains, thus, an intriguing question arises: **How can neural networks extrapolate truly "novel" domains and achieve OOD generalization?**
We introduce a novel approach to domain extrapolation that leverages reasoning ability and the extensive knowledge encapsulated within large language models (LLMs) to synthesize entirely new domains. Starting with the class of interest, we query the LLMs to extract relevant knowledge for these novel domains. We then bridge the gap between the text-centric knowledge derived from LLMs and the pixel input space of the model using text-to-image generation techniques. By augmenting the training set of domain generalization datasets with high-fidelity, photo-realistic images of these new domains, we achieve significant improvements over all existing methods, as demonstrated in both single and multi-domain generalization across various benchmarks.
With the ability to extrapolate any domains for any class, our method has the potential to learn a generalized model for any task without any data. To illustrate, we put forth a much more difficult setting termed, **data-free domain generalization**, that aims to learn a generalized model in the absence of any collected data. Our empirical findings support the above argument and our methods exhibit commendable performance in this setting, approximating the supervised with synthetic data only and even surpassing the supervised setting by approximately 1-2% on datasets such as VLCS.

## 1 INTRODUCTION

Deep neural networks have demonstrated remarkable achievements in various fields and applications He et al. (2015); Devlin et al. (2018); Chen et al. (2021); Dosovitskiy et al. (2021); Li et al. (2021), yet their effectiveness heavily depends on the assumption that the training and testing environments are subject to independent and identically distributions Ben-David et al. (2010); Blanchard et al. (2011). Out-of-distribution (OOD) generalization aims to learn model from some training distribution that can generalize well to unseen testing domains, usually with distribution or label shifts Liu et al. (2021). A typical scenario is domain generalization (DG) where multiple source domains are available and these available source domains aid the training of generalizable models that learn invariant features and discard spurious ones. However, a significant challenge arises: the availability of these source domains often becomes a limiting factor, hindering the success of current OOD approaches in more challenging scenarios Qiao et al. (2020); Wang et al. (2021); Xu et al. (2020); Wang et al. (2022), which can be attributed to the difficulty and high expenses to collect, not just, data but data in diverse domains with annotations, which is sometimes impossible in critical areas such as healthcare or extreme conditions (e.g. deep sea or space). Motivated by these challenges, domain augmentation is straightforward and multiple methods have been proposed to generate novel

domains and images through mixup Yan et al. (2020), mixing of statistics Zhou et al. (2021), uncertainty modeling Li et al. (2022b); Zhou & Konukoglu (2023) and convex combination Albuquerque et al. (2019). However, these methods generally interpolate the existing training domains to generate novel domains that still fall within the convex hall of available domains Albuquerque et al. (2019). Consequently, the constrained number of source domains hampers the expressiveness of these methods, continuing to act as a performance bottleneck. On the other hand, Humans harness the innate ability of the human brain to create novel domains as illustrated in Shu et al. (2023); Radford et al. (2021) where a pre-defined set of novel domains and styles are utilized. However, this also requires human labor which fails to scale to larger sizes. Naturally, an intriguing question arises: **How can neural networks extrapolate truly "novel" domains and achieve OOD generalization?**

Large language models (LLMs) Brown et al. (2020) have been shown to encapsulate a vast wealth of knowledge and simulate human cognitive processes. Thus, a pertinent question emerges: Can one harness the power of LLMs to produce novel domains and relevant knowledge, thereby replacing the human in the above training process? Stemming from this primary query, we investigate how we can extract knowledge of a specific task and produce novel domains from LLMs. A subsequent research question is: How can we leverage this text-centric knowledge from LLMs to instruct an image system that processes pixel input? State-of-the-art text-to-image generation models such as Imagen Saharia et al. (2022), Stable Diffusion Rombach et al. (2022b) and GLIDE Nichol et al. (2021) exhibit promising capability to synthesize photo-realistic images positioning them as the optimal conduit between textual and visual realms. Finally, we seek to answer to what extent the synthesized images based on knowledge can serve as Out-of-distribution learners that can generalize to unseen testing domains. Following these problems, we are the first study to design a new paradigm that leverages the knowledge of LLMs to extrapolate novel domains for training better generalizable and sample-efficient models. With the ability to extrapolate any domains for any class, our method has the potential to learn a generalized model for any task without any existing data.

In addition, we present **data-free domain generalization**. Data-free generalization endeavors to enable a model across unseen testing domains based solely on task specifications (for example, distinguishing between dog and cat classes) without the need for gathering or utilizing any pre-existing datasets. In the era of large foundation models, data-free domain generalization is formulated as OOD problem with inaccessible meta distribution and domain distribution (detailed in Section 2.1) – essentially, devoid of any real-world data. This scenario presents a significantly more complex challenge than that encountered in multi-domain or single-domain generalization efforts. Moreover, it holds pragmatic significance in democratizing machine learning, by urging the community to develop methodologies that are viable under stringent resource constraints. Such an initiative paves the way for wider access to and application of, machine learning. Our method not only addresses the challenge of data scarcity in DG problems but also underscores the potential of synthetic data in overcoming traditional barriers to machine learning implementation.

Extensive experiments on single, multi-domain and data-free evaluations demonstrate the effectiveness of our proposed method. In both single and multi-domain configurations, we demonstrate that synthetic data in the extrapolated novel domains markedly outperforms baseline results across various datasets. On the more challenging data-free setting, our proposed method exhibits near-supervised performance in this setting, even surpassing the supervised baseline by approximately 1-2% on VLCS. Data synthesized via the knowledge from LLMs excels compared to the synthetic data directly generated from text-to-image generation models. This demonstrates the ability of LLMs to effectively extrapolate like humans and integrate this prior knowledge into the model.

We also underscore the scalability of our approach by highlighting that as the number of domains escalates, the performance correspondingly improves. Intriguingly, this trend diverges from the outcomes observed when augmenting with synthetic data directly produced by text-to-image models reported in Azizi et al. (2023); He et al. (2022). This further demonstrates the pivotal role of the knowledge derived from LLMs in mitigating overfitting to synthetic data.

The remainder of this paper is organized as follows: In Section 2, we will first motivate our method from the perspective of the theoretical error bound for out-of-distribution (OOD) generalization. Then we will detail our method design and specifications. Section 3 introduces the data-free generalization and its potential usage in the era of large foundation models. Section 4 describes our experiment design, results and the implications of our findings. Section 5 introduces related work. Section 6 concludes our paper and potential limitation of our work.

## 2  METHOD

We motivate our method from the perspective of the theoretical error bound for OOD generalization. We will first provide the notation for the theoretical framework. Then we motivate our research problem from the OOD generalization error bound, i.e. limited number of source domains leading to a larger error bound. Then we propose a proxy method that approximates the meta-distribution with a proxy distribution. We give a new error bound on this method. Lastly, we propose one realization of our method by using LLMs to approximate the meta-distribution and text-to-image generation models to bridge the text-centric knowledge with the input pixel space.

### 2.1  THEORETICAL BOUND

**Notation.** Let $\mathcal{X}$ denote the observation space and $\mathcal{Y} = \{1, -1\}$ the output space. Denote $P_{XY}$ as the joint probability of the joint space of $\mathcal{X} \times \mathcal{Y}$ and assume a meta distribution $\mu$ and n domains $P_{XY}^{(1)}, \cdots, P_{XY}^{(i)}, P_{XY}^{(n)}$ are i.i.d realizations from $\mu$. A decision function is a function $f \in \mathcal{F} : \mathcal{X} \to \mathcal{Y}$ predicts $\hat{y}_i = f(x_i)$. We denote $l : \mathcal{Y} \times \mathcal{Y} \to \mathbb{R}_+$ a loss function and define the generalization error of a decision function as

$$\mathcal{L}^{\mu}(f) = \mathbb{E}_{P_{XY} \sim \mu} \mathbb{E}_{(x,y) \sim P_{XY}}[l(f(x), y)] \tag{1}$$

Since we have no access to $\mu$ and all the realizations $P_{XY}^{(1)}, \cdots, P_{XY}^{(i)}, P_{XY}^{(n)}$ but sampled images from these realizations, we can derive an empirical error:

$$\hat{\mathcal{L}}^{\mu}(f) = \sum_{i=1}^{n} \sum_{j=1}^{m} l(f(x_{ij}, y_{ij}) \tag{2}$$

where $(x_{ij}, y_{ij}) \sim P_{XY}^{(j)}$ denotes the $i$th sample drawn from $P_{XY}^{(j)}$. It's easy to see that when $n \to \infty, m \to \infty$, $\hat{\mathcal{L}}^{\mu}(f)$ converges to $\mathcal{L}^{\mu}(f)$, which gives the intuitive sense that increasing $m$ and $n$ gives us better-approximated solutions. This motivates us to increase $n$ and $m$, i.e.increasing the number of domains and training images per domain, which is difficult due to the inaccessible $\mu$ and $P_{XY}^{(1)}, \cdots, P_{XY}^{(i)}, P_{XY}^{(n)}$. Prior arts have proposed various methods to generate novel domains but the majority falls in the interpolation of existing domains, failing to effectively increase $n$. How can to approach this problem? **We can approximate $\mu$ by new distribution $\mu'$ sufficiently close to $\mu$ that can be sampled.**

**Definition 1** *We define the distance between the two distributions as*

$$D(\mu, \mu') = \sup_{f \in \mathcal{F}} |\mathcal{L}^{\mu'}(f) - \mathcal{L}^{\mu}(f)|$$

With the following assumption,

**Assumption 1** *We assume the distance $D(\mu, \mu') \leq \epsilon$.*

we can derive a bound through the approximated $\mu'$.

**Theorem 1** *With confidence at least $1 - 2\delta$ and for all $f \in \mathcal{F}$, we have*

$$\mathcal{L}^{\mu}(f) \leq \hat{\mathcal{L}}^{\mu'}(f) + 2\mathcal{R}_{mn}(\mathcal{F}) + 2\mathcal{R}_{n}(\mathcal{F}) + 3\sqrt{\frac{\ln(2/\delta)}{2mn}} + 3\sqrt{\frac{\ln(2/\delta)}{n}} + \epsilon$$

Proof in Appendix A. By replacing $\mu$ with $\mu'$, we now have control over $\hat{\mathcal{L}}^{\mu'}(f)$, $m$ and $n$ as we can sample as many domains and images from $\mu'$ as possible. This is obtained at the cost of $\epsilon$, which we assume to be small.

**Remark 1** *We also note that as $n$ and $m$ increase, the upper bound of the generalization error decreases, which gives us better generalization errors.*

With sufficiently large $n$ and $m$, the decrease part of the generalization error will cancel out the cost of $\epsilon$, leading to a lower generalization error.

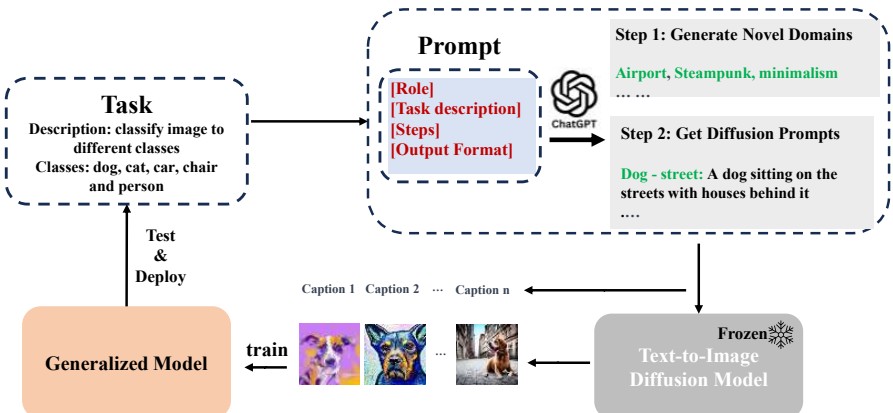

Figure 1: Overall pipeline of our paradigm: *Extrapolation of novel domains via the knowledge of LLMs*, a novel learning paradigm where knowledge from LLMs assists the training of generalizable models via text-to-image models in a completely data-free fashion.

## 2.2 DOMAIN EXTRAPOLATION WITH LLMS

Given the aforementioned theoretical bound, our objective is to approximate $\mu$ with $\mu'$. Humans, as evidenced in Shu et al. (2023); Radford et al. (2021), usually can efficiently extrapolate novel domains (by imagination), which is a good approximation of $\mu$. Nonetheless, human intervention is expensive and not scalable to arbitrary datasets. Conversely, LLMs not only embody a vast expanse of knowledge Petroni et al. (2019) and exhibit comparable reasoning capabilities Qiao et al. (2023), but they also present the benefit of being amenable to extensive sampling. To this end, we propose to query LLMs, in place of human, to extrapolate novel domains.

After sampling from meta distribution $\mu'$, we need to further sample from the domain distribution to generate images in this particular novel domain. As discussed in Section 1, this leads to a gap between the text-based knowledge output by the LLMs and the input pixel space of vision systems. Text-to-image generation models (e.g. stable diffusion Rombach et al. (2022a)) exhibit the great capability to output photo-realistic images through inputting texts positioning them as the optimal bridge between textual and visual realms. The synthetic images of extrapolated novel domains are used to augment the original dataset or train the models solely in a data-free fashion. An overall illustration of our paradigm can be seen in Figure 1.

**Extracting Knowledge from LLMs.** The objective is to approximate $\mu$ via LLMs as close as possible. This introduces a constraint whereby the generated novel domains must reside within the high-density regions of distribution $\mu$. To ensure adherence to this criterion, we purposefully instruct the LLMs to conceive the most plausible and reasonable domains where a particular class would realistically exist. To better guide LLMs to understand the instruction and generate the required response accordingly, we craft system prompts that include role description ([Role]) and task description ([Task Description]), as illustrated by the example in Figure 2. Numerous strategies exist to solicit knowledge and novel domains from LLMs.

*Dataset-wise query.* The most direct approach entails querying the LLMs with comprehensive dataset information (i.e. all of the class names) and instructing the model to produce $n$ novel domains. However, as the marginal distribution for each class might exhibit minimal overlap (worse when the number of classes grows), it becomes considerably intricate to sample novel domains that are both plausible and likely for all classes.

*Class-wise query.* Thus, we propose to query the LLMs for novel domains of specific classes. For each class in the task, we query the LLMs for knowledge and $n$ novel domain information specific to that class. We repeat the process one class after another until all of the classes are iterated. We provide a example prompt in Figure 2.

**Bridging text and pixel with text-to-image generation models.** After obtaining a number of the most plausible and reasonable domains of a specific class, we transform the text-centric knowledge from LLMs to pixel space by text-to-image generation models. This process is exactly the real-

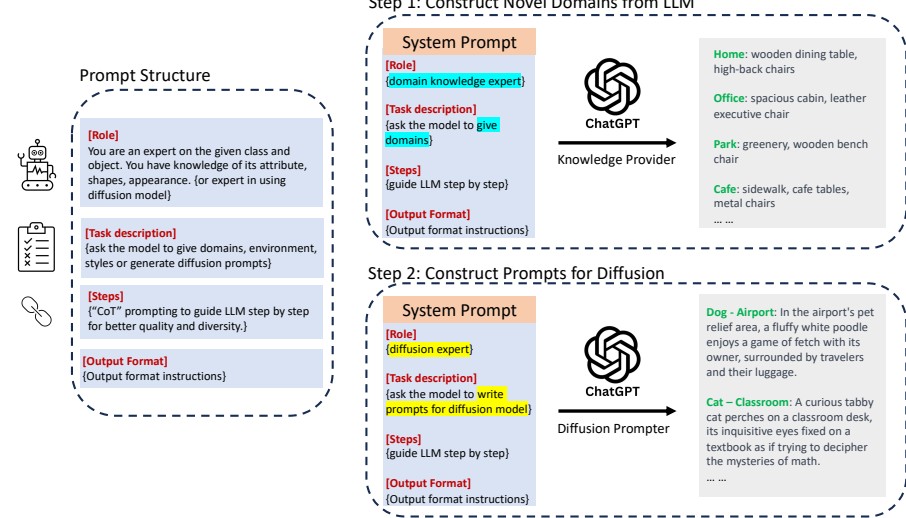

Figure 2: Knowledge extraction pipeline. We first employ various SOTA prompting methods: e.g. "Chain of Thought Wei et al. (2022)" (CoT) prompting, role prompting to extract domains from LLM (Step 1) and automatically generate prompt for a Text-to-Image model. (Step 2)

ization of sampling $X$ from $P_X^{(i)}$ where $P_X^{(i)}$ is the $i$th domain generated by $\mu'$ (i.e. the LLM). Numerous strategies exist to prompt text-to-image generation models conditioned on class and domain information.

*Template prompt.* The most immediate strategy involves employing templates as prompts (e.g., "an image of [CLASS_NAME] in the domain of [DOMAIN_NAME]"). However, the limitation lies in its lack of diversity: utilizing the identical prompt to produce multiple images results in images bearing resemblance to one another.

*LLM generated prompt.* Thus, we propose to query the LLMs for prompts conditioned on the class name and domain information acquired in the previous step. As illustrated in Figure 2, we craft system prompts that specifically tailor the LLM to generate prompts for text-to-image generation models and generate multiple prompts for each of the novel domains of each class.

## 3 DATA-FREE DOMAIN GENERALIZATION

We present Data-free Generalization, a new formation of generalization in the era of large foundation models. Given a task with detailed description and requirements (e.g. the classes to be classified and the definition of each class), Data-free Generalization endeavors to learn a model that can generalize to this specific task and fulfill the requirement without collecting any data or utilizing any existing datasets. Formally, this problem is formulated as follows. Task description and requirements specify the decision function $f \in \mathcal{F} : \mathcal{X} \to \mathcal{Y}$ and the meta distribution $\mu$. The problem then turns to minimizing Equation 3, as detailed in Section 2.1. The difference is that now the meta distribution $\mu$ cannot be sampled and thus we have no access to any training domains $P_{XY}^{(1)}, \cdots, P_{XY}^{(i)}, P_{XY}^{(n)}$ or images that are sampled from these domains. However, in the era of large foundation models, the meta distribution $\mu$ can be approximated by LLMs while the domain distribution can be approximated by image generation models. Consequently, we can provide a guarantee on the learning with Theorem 1. We provide one such method in Section 2.

Data-free generalization can not only serve a more difficult setting to push the limits of current OOD methods but also holds pragmatic significance in democratizing machine learning. It does so by mitigating or potentially eliminating the necessity for data collection and annotation within the machine learning pipeline, which facilitates a broader access to and application of machine learning technologies, particularly for entities facing resource constraints. Envision a modest-sized enterprise incapable of investing in the training of large foundational models, nor possessing the necessary time and funding to collect and label an extensive dataset for particular tasks. This situation aligns with

the concept of Data-free Generalization, characterized by the availability of only task specifications in the absence of accessible data. Our methodology offers an ideal resolution for such organizations. Initially, they can leverage LLMs' APIs for a limited number of queries to derive extrapolated domains and scenarios. Following this, they may engage text-to-image models for data synthesis. This synthetic data can then be utilized to either develop new models or enhance existing ones, thereby circumventing the limitations posed by resource constraints.

## 4 EXPERIMENTS

The objective of our experiments is to (i) demonstrate that knowledge from LLMs successfully extrapolates novel domains and leads to performance benefits grounded by theoretical bounds. (ii) Investigate the most efficient and effective approach for extracting knowledge and sampling from text-to-image models. (iii) Analyze to what extent the synthetic images generated condition on LLMs' knowledge can serve as good out-of-distribution learners that lead to generalization on unseen testing domains.

### 4.1 EXPERIMENT SETUP

**Setup.** OOD Generalization is evaluated on DomainBed Gulrajani & Lopez-Paz (2020) with four datasets, i.e. PACS, VLCS, OfficeHome and DomainNet and we follow them on the the train-validate-test split of each dataset to perform the hyperparameter search. For comprehensive evaluation, we experiment on both multi- and single-domain generalization protocols. In addition, we propose the data-free domain generalization to evaluate training generalizable models in a data-free fashion with only task information.

**Baseline.** We set two baselines for our experiments, namely empirical risk minimization (ERM) and ERM with exponential moving average (ERM + EMA). ERM with EMA is demonstrated to be more stable and effective than ERM Arpit et al. (2022). It is thus adopted to perform ablation study and analysis.

**Implementation.** We remove the dropout and follow the rest of the implementation as in Gulrajani & Lopez-Paz (2020) since dropout is reported to harm some of the DG methods Huang et al. (2022), e.g. RSC Huang et al. (2020). We adopt GPT-4 to extract novel domain knowledge and leverage Stable Diffusion 2 Rombach et al. (2021) as the text-to-image generation model. We use one A100 GPU to generate synthetic images. All experiments of training ResNet50 and CLIP ViT-B16 model can be run on 1 RTX3090 GPU.

### 4.2 MAIN RESULTS

**Leave-one-out evaluation.** Leave-one-out Evaluation leaves one domain as the testing domain and uses the rest as training domains. For our method, all of the synthetic images are treated as an additional domain to the source domains. As per Table 1, augmenting with the novel domain synthetic images leads to a consistent improvement (as large as 5.2%) over the ERM and ERM + EMA baselines. On average, we achieve a 2.9% and 2.4% improvement over ERM and ERM + EMA baselines respectively. Our method also achieved a significant improvement (1.2% on average) over the CLIP fine-tuned baseline. This improvement is remarkable, given the already high performance of the CLIP model. In addition to the CLIP baseline, we also compare with SOTA methods that adopts CLIP as the backbone. It's noteworthy that DCLIP Menon & Vondrick (2022) and WaffleCLIP Roth et al. (2023) also utilize the knowledge of LLMs to boost performance. Among these SOTAs, our method still achieves the best Averaged result, bypassing the second best by more than 1%.

**Single Domain Generalization.** Single-domain generalization Evaluation leverages a single domain for training and subsequently assesses the outcomes on the remaining domains. This scenario presents a greater challenge when juxtaposed with the Leave-one-out setting due to the model's exclusive exposure to just one domain during its training phase. Such a setting accentuates the issue of restricted availability of source domains. Considering our methodology does not impose assumptions on either the source domains or the model, but instead extrapolates novel domains via LLMs to augment the training set, it is optimally more suited for this specific context. Empirical evidence underscores its exceptional efficacy and with merely one source domain of real images, our results

Table 1: Leave-one-out Evaluation on DomainBed. CLIP adopts ViT-B16 as the backbone. † denotes reproduced results. MixStyle result is taken from Cha et al. (2021b)

| Algorithm | VLCS | PACS | OfficeHome | DomainNet | Avg |
|---|---|---|---|---|---|
| ERM Vapnik (1998) | 77.5 ± 0.4 | 85.5 ± 0.2 | 66.5 ± 0.3 | 40.9 ± 0.1 | 67.6 |
| IRM Arjovsky et al. (2019) | 78.5 ± 0.5 | 83.5 ± 0.8 | 64.3 ± 2.2 | 33.9 ± 2.8 | 65.1 |
| GroupDRO Sagawa et al. (2019) | 76.7 ± 0.6 | 84.4 ± 0.8 | 66.0 ± 0.7 | 33.3 ± 0.2 | 65.1 |
| MLDG Li et al. (2018a) | 77.2 ± 0.4 | 84.9 ± 1.0 | 66.8 ± 0.6 | 41.2 ± 0.1 | 67.5 |
| CORAL Sun & Saenko (2016) | 78.8 ± 0.6 | 86.2 ± 0.3 | 68.7 ± 0.3 | 41.5 ± 0.1 | 68.8 |
| Mixup Yan et al. (2020) | 78.1 ± 0.3 | 86.8 ± 0.3 | 68.0 ± 0.2 | 39.6 ± 0.1 | 68.1 |
| MMD Li et al. (2018b) | 77.9 ± 0.1 | 87.2 ± 0.1 | 66.2 ± 0.3 | 23.5 ± 9.4 | 63.7 |
| RSC Huang et al. (2020) | 77.8 ± 0.6 | 86.2 ± 0.5 | 66.5 ± 0.6 | 38.9 ± 0.6 | 67.4 |
| VREx Krueger et al. (2021) | 78.1 ± 0.2 | 87.2 ± 0.6 | 65.7 ± 0.3 | 30.1 ± 3.7 | 65.3 |
| SWAD Cha et al. (2021b) | 79.1 ± 0.4 | 88.1 ± 0.4 | 70.6 ± 0.3 | 46.5 ± 0.2 | 66.9 |
| MIRO Cha et al. (2022) | 79.0 ± 0.2 | 85.4 ± 0.4 | 70.5 ± 0.4 | 44.3 ± 0.2 | 65.9 |
| MixStyle Zhou et al. (2021) | 77.9 | 85.2 | 60.4 | 34.0 | 64.4 |
| RISE Huang et al. (2023) | 81.7 | 89.4 | 71.6 | - | - |
| EoA Arpit et al. (2022) | 79.1 | 88.6 | 72.5 | 47.4 | 71.9 |
| StyleNeophile Kang et al. (2022) | - | 89.1 | 65.9 | 44.6 | - |
| XDED Lee et al. (2022) | 74.8 | 83.8 | 65.0 | - | - |
| ERM † Vapnik (1998) | 77.2 ± 1.0 | 84.4 ± 0.8 | 64.8 ± 0.4 | 43.6 ± 0.1 | 67.5 |
| + ours | 78.5 ± 0.4 | 88.0 ± 0.3 | 70.0 ± 0.1 | 45.2 ± 0.1 | 70.4 |
| Δ | + 1.3 | + 3.6 | + 5.2 | + 1.6 | + 2.9 |
| ERM + EMA | 78.8 ± 0.6 | 87.8 ± 0.3 | 70.5 ± 0.1 | 46.0 ± 0.1 | 70.8 |
| + ours | 80.2 ± 0.3 | 90.3 ± 0.4 | 74.6 ± 0.2 | 47.5 ± 0.3 | 73.2 |
| Δ | + 1.4 | + 2.5 | + 4.1 | + 1.5 | + 2.4 |
| CLIP Zero-shot | 80.1 | 96.2 | 83.0 | 58.5 | 79.5 |
| CLIP Finetune | 82.4 ± 0.1 | 95.3 ± 0.2 | 84.8 ± 0.1 | 59.9 ± 0.1 | 80.6 |
| PromptStyler Cho et al. (2023) | 82.9 | 97.2 | 83.6 | 59.4 | 80.8 |
| DCLIP Menon & Vondrick (2022) | 81.2 | 96.6 | 82.6 | 57.3 | 79.4 |
| WaffleCLIP Roth et al. (2023) | 83.1 | 96.4 | 82.3 | 59.1 | 80.2 |
| + concepts | 81.8 | 96.4 | 83.8 | 60.1 | 80.5 |
| + ours | 82.7 ± 0.3 | 96.5 ± 0.3 | 86.5 ± 0.2 | 61.3 ± 0.0 | 81.8 |
| Δ | + 0.3 | + 1.2 | + 1.7 | + 1.4 | + 1.2 |

Table 2: Single-domain Evaluation on DomainBed. CLIP adopts ViT-B16 as the backbone.

| Algorithm | VLCS | PACS | OfficeHome | Avg |
|---|---|---|---|---|
| ASA Fan et al. (2021) | - | 67.0 | - | - |
| Pro-RandConv Choi et al. (2023) | - | 67.0 | - | - |
| CPerb Zhao et al. (2023) | - | 73.3 | - | - |
| RSC Huang et al. (2020) | 59.2 ± 0.7 | 60.9 ± 1.7 | 46.9 ± 1.7 | 55.7 |
| ERM (Multi-domain) | 77.2 ± 1.0 | 84.4 ± 0.8 | 64.8 ± 0.4 | 75.5 |
| ERM Vapnik (1998) | 59.2 ± 0.8 | 64.6 ± 0.6 | 51.5 ± 0.3 | 58.4 |
| + ours | 76.3 ± 0.2 | 83.9 ± 0.9 | 64.7 ± 0.2 | 75.0 |
| Δ | + 17.1 | + 19.3 | + 13.2 | + 16.5 |
| ERM + EMA (Multi-domain) | 78.8 ± 0.6 | 87.8 ± 0.3 | 70.5 ± 0.1 | 79.0 |
| ERM + EMA | 64.2 ± 0.7 | 67.9 ± 1.1 | 58.2 ± 0.1 | 62.7 |
| + ours | 78.0 ± 0.1 | 87.6 ± 0.6 | 69.4 ± 0.3 | 78.3 |
| Δ | +13.1 | +21.7 | +12.0 | +15.6 |

closely mirror, and at times even surpass, those obtained in a multi-domain configuration, as per Table 2. Specifically, we achieve the highest of 78.0%, 87.6%, 69.4% on the three datasets, outperforming the ERM with multiple source domains by margins of 0.8%, 3.2% and 4.6% respectively. Compared to baselines, our method achieves a remarkable improvement of over 10% across all datasets and baselines. This evidences that our methodology substantially mitigates the challenges associated with restricted source domains, rendering it particularly optimal and effective in scenarios where source domains are unavailable, such as single-domain generalization.

**Comparison with augmentation-based DG methods.** We compared with SOTA augmentation methods in Table 4 including MixStyle Zhou et al. (2021), DSU Li et al. (2022b), AutoAug Cubuk et al. (2018) and RandAug Cubuk et al. (2020), where our method demonstrates an improvement of more than 2% on average.

| Algorithm | VLCS | PACS | OfficeHome | DomainNet | Avg |
|---|---|---|---|---|---|
| ERM | | | | | |
| Multi-domain | 77.2 ± 1.0 | 84.4 ± 0.8 | 64.8 ± 0.4 | 43.6 ± 0.1 | 67.5 |
| Single-domain | 59.2 ± 0.8 | 64.6 ± 0.6 | 51.5 ± 0.3 | - | - |
| Data-free (ours) | 73.9 ± 0.3 | 82.5 ± 0.9 | 62.1 ± 0.1 | 25.9 ± 0.2 | 61.1 |
| ERM + EMA | | | | | |
| Multi-domain | 78.8 ± 0.6 | 87.8 ± 0.3 | 70.5 ± 0.1 | 46.0 ± 0.1 | 70.8 |
| Single-domain | 64.2 ± 0.7 | 67.9 ± 1.1 | 58.2 ± 0.1 | - | - |
| Data-free (ours) | 79.9 ± 0.6 | 86.9 ± 0.1 | 67.4 ± 0.2 | 30.3 ± 0.1 | 66.1 |

Table 3: Data-free generalization on DomainBed.

| Algorithm | VLCS | PACS | Avg |
|---|---|---|---|
| MixStyle Zhou et al. (2021) | 78.7 ± 0.1 | 87.7 ± 0.1 | 83.2 |
| DSU Li et al. (2022b) | 77.7 ± 0.0 | 87.6 ± 0.2 | 82.7 |
| AutoAug Cubuk et al. (2018) | 78.6 ± 0.3 | 88.6 ± 0.1 | 83.6 |
| RandAug Cubuk et al. (2020) | 79.1 ± 0.0 | 87.5 ± 0.3 | 83.3 |
| ERM + EMA | 78.8 ± 0.6 | 87.8 ± 0.3 | 83.3 |
| +larger batch-size | 78.1 ± 0.1 | 87.4 ± 0.1 | 82.7 |
| + class-template | 79.3 ± 0.1 | 88.0 ± 0.3 | 83.7 |
| + class-prompt | 79.3 ± 0.0 | 88.5 ± 0.2 | 83.9 |
| + ours | **80.2 ± 0.3** | **90.3 ± 0.4** | **85.3** |

Table 4: Comparison with two baselines and current SOTA augmentation-based DG methods. All models are equipped with EMA for fair comparison.

## 4.3 DATA-FREE GENERALIZATION.

Data-free Generalization Evaluation serves as a more difficult setting to evaluate our proposed methods. Data-free Generalization aims to generalize to unseen testing domains with only knowledge of the task, i.e. the classes and definition of each class are available and no available data of any kind. To simulate Data-free Generalization with existing benchmarks, we use all the domains in existing DG datasets as testing domains. To evaluate our method, we directly train models on the synthetic images generated conditioned on novel domain knowledge. Then the model is tested on all the available real images of the domains for evaluation. Results are illustrated in Table 3 where we achieve the highest performance of 79.9%, 86.9%, 67.4% with only less than 1% gap between its multi-domain counterparts and largely surpasses single-domain counterparts. Notably, data-free ERM + EMA presents an accuracy of 79.9% on VLCS outperforming the multi-domain supervised counterparts by more than 1%. With the knowledge injected and novel domain extrapolated, this empirical result illustrates the promise of achieving generalization in a completely data-free fashion free of laborious data collection and annotation.

## 4.4 ABLATION STUDY AND ANALYSIS

To fully understand the performance of our method, we perform an ablation study by first providing three baselines building upon ERM + EMA with minor modifications. First, we provide **larger batchsize** baseline, which is used to ablate the influence of larger batch sizes incurred by the additional augmentation data. Then, we provide **class template** baseline, which prompts the text-to-images generation model to generate synthetic images with the template "An image of [CLASS]". Then we will provide a third baseline, termed **class prompt** that will prompt LLMs to give a diffusion-style prompt (without explicitly instructing it to extrapolate novel domains) and use the generated prompts to query text-to-image models for synthetic data. Comparison is shown in Table 4. We can see that a larger batch size in fact has a negative effect while both template and prompt baseline underperform our method. This ablates the influence brought by text-to-image models and further underscores the importance of LLMs' knowledge regarding the novel domain.

**Comparison between different knowledge extraction.** We provide three approaches to extract knowledge regarding the novel domains of particular classes. Comparison can be seen in (b) of Figure 4, where we show that, overall, class-wise combined with LLM-generated prompt leads to better performance than class-wise query only and data-wise query. This is because class-wise query provides more plausible and reasonable novel domains given some class and LLM-generated prompt further extracts knowledge regarding this novel domain and increases diversity in generation.

**Scaling to larger synthetic dataset.** It has been widely reported that data generated by generation models negatively impacts the model, especially when the number of synthetic images grows at scale He et al. (2022); Azizi et al. (2023). To this end, we investigate whether the performance increases scales with more synthetic data from more extrapolated novel domains. We perform scaling by prompting LLMs to extrapolate more novel domains and generate 64 image per domain. We can see in Figure 3 that with more domains (larger $n$ in Section 2.1), performance keeps increasing, which is consistent with our theoretical framework. We also make a comparison with class-template and class-prompt baselines and scale the two baselines by increasing the synthetic images to the corresponding size. However, these two methods both suffer from performance saturation and degradation when synthetic data increases, which is consistent with previous studies He et al. (2022); Azizi

et al. (2023). This demonstrated that our method can scale better to larger sizes of synthetic data and underscore the importance of new knowledge injected by LLMs that benefits generalization.

| variance measure on | PACS |
|---|---|
| LLMs extrapolation | $89.87 \pm 0.4$ |
| text-to-image generation | $89.72 \pm 0.2$ |
| model training | $90.3 \pm 0.4$ |

Table 5: Variance analysis over the three modules to measure how stable our method performs.

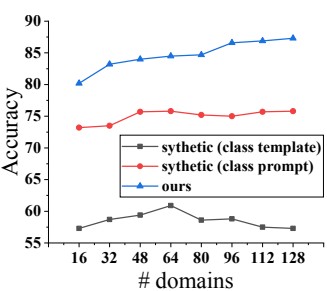

Figure 3: Scaling the training dataset by adding more novel domains. Each novel domain consists of 64 images. To facilitate fair comparison, we scale the class template method by the same amount of images.

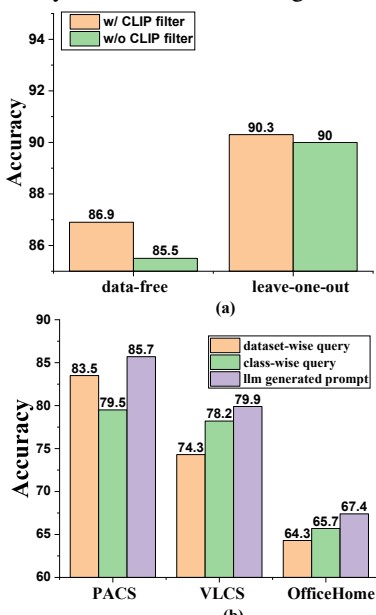

Figure 4: (a) Effectiveness of CLIP filtering. (b) Comparison between different knowledge extraction methods.

**Variance Analysis.** We aim to measure how stable our method is by decomposing the variance into three parts, i.e. LLMs extrapolation, text-to-image generation and final model training. We repeat each experiment three times and report the average and standard deviation in Table 5. For instance, to conduct variance anlysis on text-to-image generation, we use the same set of novel domains generated by LLMs, can generate synthetic datasets with the same text-to-image model three times. As per the table, we can see that all three parts contribute to a relatively small variance, suggesting that our method is stable.

**Additional CLIP filtering.** Text-to-image generation models are essentially noisy and might generate images of distortion or without the main class of interest. We experiment with CLIP filtering before the training process. As shown in (a) of Figure 4, we can observe an increase with additional filtering techniques by 1 %. To further illustrate the effectiveness of filtering, we visualize some filtered failure cases in Appendix C.

**Different LLMs.** To make sure that our method does not reply on specific LLMs, i.e. ChatGPT-4, we conduct experiments with LLMs from different families, e.g Llama and Mixtral in table .

| LLM | A | C | P | S | Avg |
|---|---|---|---|---|---|
| GPT-4 | $94.4 \pm 0.2$ | $85.0 \pm 0.5$ | $98.5 \pm 0.1$ | $83.3 \pm 1.7$ | 90.3 |
| Llama-13B | $92.6 \pm 0.5$ | $83.2 \pm 0.5$ | $98.2 \pm 0.1$ | $80.9 \pm 0.7$ | 88.7 |
| Llama-70B | $93.0 \pm 0.4$ | $83.6 \pm 0.4$ | $98.5 \pm 0.2$ | $81.9 \pm 0.4$ | 89.3 |
| Mixtral-8x7B | $92.4 \pm 0.0$ | $84.6 \pm 0.3$ | $98.8 \pm 0.0$ | $81.1 \pm 0.6$ | 89.2 |

Table 6: Performance with different LLMs.

**Visualization.** We provide visualization to validate that our method do extrapolate novel domains and generate the desired class. We demonstrate generated images from three different novel domains of the PACS dataset in the last four columns of Figure 5 and compare them with the real images in the PACS dataset (first two columns). We can see that the generated novel domains are by no means an interpolation of the real domains and are different from the existing training domains by a large margin. This illustrates that our method takes one step further toward "truly" extrapolation of novel domains without human labor. We provide more visualization in the Appendix.

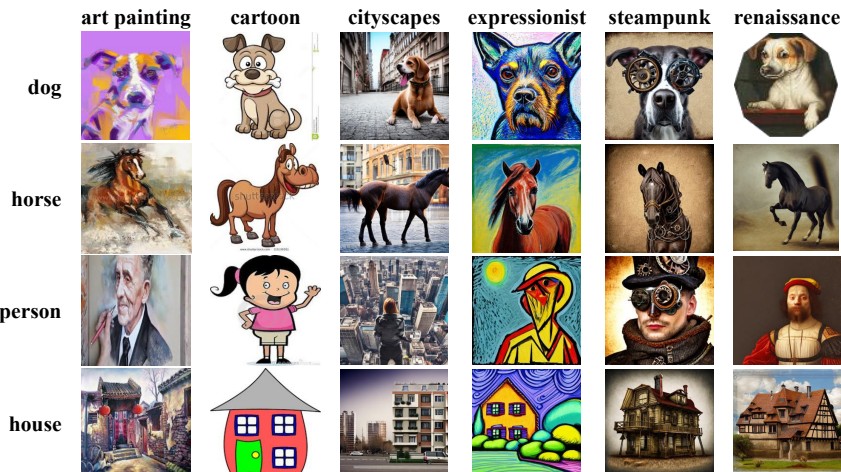

Figure 5: Examples of synthetic images conditioned on novel domain knowledge from LLM. The first two columns (i.e. art painting and cartoon) are selected from PACS datasets while the rest four columns are images generated based on the novel domains (i.e. cityscapes, etc) provided by LLMs.

## 5 RELATED WORK

**Domain Generalization.** Various approaches have been proposed to solve this problem, such as domain alignment Li et al. (2018b;c), meta-learning Li et al. (2018a); Balaji et al. (2018), ensemble learning Cha et al. (2021a); Arpit et al. (2022) and augmentation-based Zhou & Konukoglu (2023); Zhou et al. (2021); Li et al. (2022b); Xu et al. (2020); Zhou et al. (2020); Albuquerque et al. (2019). Augmentation-based methods are closely related to this work, both with the intention of generating more source domains to approximate the expected generalization error. However, these methods resort to interpolation of existing domains and fail to extrapolate the "truly" novel domains. For instance, MixStyle Zhou et al. (2021) mixes the statistics of two samples by linear interpolation. More recently, with the advent of vision-language models such as CLIP Radford et al. (2021) and Stable Diffusion Rombach et al. (2021), researchers propose to utilize Stable Diffusion to identify and cure shortcuts Wu et al. (2023) or CLIP to generate novel domain augmentation Vidit et al. (2023). However, they all require some form of human labor to pre-define a set of domains or styles, which makes them laborious and not scalable. Our work aims to solve this problem and achieve genuine domain extrapolation.

**Language scaffolded vision** aims to develop better and more robust vision systems with the help of language. Our method also falls within this category. Clipood Shu et al. (2023) proposes to fine-tune a CLIP model to adapt the downstream DG tasks by a text similarity aware loss. Min et al. (2022) utilize an RNN as an explanation network enforcing the model to self-explain, thereby increasing the robustness. Yang et al. (2023) utilize language models to produce a comprehensive set of bottleneck features and leverage CLIP to classify. With the help from LLMs, Yang et al. (2023) has pushed the performance of the bottleneck network to SOTA. Despite many works proposed, this research, to the best of our knowledge, is the first endeavor to investigate the potential of a Large Language Model (LLM) in facilitating the training of a robust and generalizable vision model.

## 6 CONCLUSION

The limited availability of domains has been a prevailing problem in Domain Generalization. In this work, we propose the first data-free learning paradigm that leverages the knowledge and reasoning of LLMs to extrapolate novel domains. By bridging the text-centric knowledge and pixel input space by sampling from text-to-image generation models, we are able to train generalizable models with task information only. Extensive experiments have demonstrated that our method achieves significant improvements over baselines and the state-of-the-art by a significant margin. We also demonstrate a promising learning paradigm where LLMs' knowledge combined with text-to-image generation models are sufficient to train a generalizable model to any task.

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
