# A  PROOF OF THEOREM 1

**Notation.** Let $\mathcal{X}$ denote the observation space and $\mathcal{Y} = \{1, -1\}$ the output space. Denote $P_{XY}$ as the joint probability of the joint space of $\mathcal{X} \times \mathcal{Y}$ and assume a meta distribution $\mu$ and n domains $P_{XY}^{(1)}, \cdots, P_{XY}^{(i)}, P_{XY}^{(n)}$ are i.i.d realizations from $\mu$. A decision function is a function $f \in \mathcal{F} : \mathcal{X} \to \mathcal{Y}$ predicts $\hat{y}_i = f(x_i)$. We denote $l : \mathcal{Y} \times \mathcal{Y} \to \mathbb{R}_+$ a loss function and define the generalization error of a decision function as

$$\mathcal{L}^\mu(f) = \mathbb{E}_{P_{XY} \sim \mu} \mathbb{E}_{(x,y) \sim P_{XY}}[l(f(x), y)] \tag{3}$$

Since we have no access to $\mu$ and all the realizations $P_{XY}^{(1)}, \cdots, P_{XY}^{(i)}, P_{XY}^{(n)}$ but sampled images from these realizations, we can derive an empirical error:

$$\hat{\mathcal{L}}^\mu(f) = \sum_{i=1}^n \sum_{j=1}^m l(f(x_{ij}), y_{ij}) \tag{4}$$

It's easy to see that when $n \to \infty, m \to \infty$, $\hat{\mathcal{L}}^\mu(f)$ converges to $\mathcal{L}^\mu(f)$, which gives the intuitive sense that increasing $m$ and $n$ gives us better-approximated solutions.

To prove Theorem 1, we use a modified version of the standard empirical Rademacher complexity bound that weakens the i.i.d assumption to an independence assumption Mohri et al. (2018).

**Theorem 2** *For distribution $P^{(1)}, \cdots, P^{(n)}$ independent sampled from meta-distribution $\mu$, and 1-Lipschitz loss $l(\cdot, \cdot)$ taking values in $[0, 1]$, the following holds with confidence at least $1 - \delta$,*

$$\frac{1}{n} \sum_{j=1}^n \mathcal{L}_{P^{(j)}}(f) \leq \frac{1}{n} \sum_{j=1}^n \hat{\mathcal{L}}_{P^{(j)}}(f) + 2\mathcal{R}_{mn}(\mathcal{F}) + 3\sqrt{\frac{\ln(2/\delta)}{2mn}} \tag{5}$$

*where $\hat{\mathcal{L}}_{P^{(j)}}(f)$ is losses on empirical set $S_{P^{(j)}}$ i.i.d. drawn from $P^{(j)}$.*

**Proof 1** *Let $S = \cup_{i=1}^n S_{P^{(i)}}$ and*

$$\Phi(S) = \sup_{f \in \mathcal{F}} \frac{1}{n} \sum_{j=1}^n (\mathcal{L}_{P^{(j)}}(f) - \hat{\mathcal{L}}_{P^{(j)}}(f)) \tag{6}$$

*which satisfies the bounded differences property required by McDiarmid's inequality, which implies that with confidence at least $1 - \frac{1}{2}\delta$ that*

$$\Phi(S) \leq \mathbb{E}_{S_{P^{(1:n)}} \sim P^{(1:n)}}[\Phi(S)] + \sqrt{\frac{\ln(2/\delta)}{2mn}} \tag{7}$$

Then we can bound the expected value of $\Phi(S)$

$$\mathbb{E}_{S_{P^{(1:n)}} \sim P^{(1:n)}} \left[ \Phi(S) \right] \tag{8}$$

$$= \mathbb{E}_{S_{P^{(1:n)}} \sim P^{(1:n)}} \left[ \sup_{f \in \mathcal{F}} \frac{1}{n} \sum_{j=1}^{n} (\mathcal{L}_{P^{(j)}}(f) - \hat{\mathcal{L}}_{P^{(j)}}(f)) \right] \tag{9}$$

$$= \mathbb{E}_{S_{P^{(1:n)}} \sim P^{(1:n)}} \left[ \sup_{f \in \mathcal{F}} \frac{1}{n} \sum_{j=1}^{n} \left( \mathbb{E}_{S'_{P^{(j)}} \sim P^{(j)}} \left[ \frac{1}{m} \sum_{i=1}^{m} l(f(x'_{ij}), y'_{ij}) \right] - \frac{1}{m} \sum_{i=1}^{m} l(f(x_{ij}), y_{ij}) \right) \right] \tag{10}$$

$$\leq \mathbb{E}_{S_{P^{(1:n)}} \sim P^{(1:n)}} \mathbb{E}_{S'_{P^{(1:n)}} \sim P^{(1:n)}} \left[ \sup_{f \in \mathcal{F}} \frac{1}{n} \sum_{j=1}^{n} \frac{1}{m} \sum_{i=1}^{m} l(f(x'_{ij}), y'_{ij}) - l(f(x_{ij}), y_{ij}) \right] \tag{11}$$

$$= \mathbb{E}_{S_{P^{(1:n)}} \sim P^{(1:n)}} \mathbb{E}_{S'_{P^{(1:n)}} \sim P^{(1:n)}} \mathbb{E}_{\sigma} \left[ \sup_{f \in \mathcal{F}} \frac{1}{n} \sum_{j=1}^{n} \frac{1}{m} \sum_{i=1}^{m} \sigma_{ij}(f(x'_{ij}), y'_{ij}) - l(f(x_{ij}), y_{ij}) \right] \tag{12}$$

$$\leq \mathbb{E}_{S'_{P^{(1:n)}} \sim P^{(1:n)}} \mathbb{E}_{\sigma} \left[ \sup_{f \in \mathcal{F}} \frac{1}{n} \sum_{j=1}^{n} \frac{1}{m} \sum_{i=1}^{m} \sigma_{ij}(f(x'_{ij}), y'_{ij}) \right] \tag{13}$$

$$+ \mathbb{E}_{S_{P^{(1:n)}} \sim P^{(1:n)}} \mathbb{E}_{\sigma} \left[ \sup_{f \in \mathcal{F}} \frac{1}{n} \sum_{j=1}^{n} \frac{1}{m} \sum_{i=1}^{m} -\sigma_{ij}(f(x_{ij}), y_{ij}) \right] \tag{14}$$

$$= 2 \mathbb{E}_{S_{P^{(1:n)}} \sim P^{(1:n)}} \mathbb{E}_{\sigma} \left[ \sup_{f \in \mathcal{F}} \frac{1}{n} \sum_{j=1}^{n} \frac{1}{m} \sum_{i=1}^{m} \sigma_{ij} l(f(x_{ij}), y_{ij}) \right] \tag{15}$$

$$= 2 \mathbb{E}_{S_{P^{(1:n)}} \sim P^{(1:n)}} \left[ \mathcal{R}_{mn}(\mathcal{F}) \right] \tag{16}$$

Following McDiarmid's inequality, we know that with confidence at least $1 - \frac{1}{2}\delta$,

$$2 \mathbb{E}_{S_{P^{(1:n)}} \sim P^{(1:n)}} \left[ \mathcal{R}_{mn}(\mathcal{F}) \right] \leq 2 \mathcal{R}_{mn}(\mathcal{F}) + 2\sqrt{\frac{\ln(2/\delta)}{2mn}} \tag{17}$$

Finally, we have

$$\Phi(S) = \sup_{f \in \mathcal{F}} \frac{1}{n} \sum_{j=1}^{n} (\mathcal{L}_{P^{(j)}}(f) - \hat{\mathcal{L}}_{P^{(j)}}(f)) \tag{18}$$

$$\leq \mathbb{E}_{S_{P^{(1:n)}} \sim P^{(1:n)}} \left[ \Phi(S) \right] + \sqrt{\frac{\ln(2/\delta)}{2mn}} \tag{19}$$

$$\leq 2 \mathcal{R}_{mn}(\mathcal{F}) + 2\sqrt{\frac{\ln(2/\delta)}{2mn}} + \sqrt{\frac{\ln(2/\delta)}{2mn}} \tag{20}$$

$$= 2 \mathcal{R}_{mn}(\mathcal{F}) + 3\sqrt{\frac{\ln(2/\delta)}{2mn}} \tag{21}$$

Thus,

$$\frac{1}{n} \sum_{j=1}^{n} \mathcal{L}_{P^{(j)}}(f) \leq \frac{1}{n} \sum_{j=1}^{n} \hat{\mathcal{L}}_{P^{(j)}}(f) + 2 \mathcal{R}_{mn}(\mathcal{F}) + 3\sqrt{\frac{\ln(2/\delta)}{2mn}} \tag{22}$$

which completes the proof.

Then we can derive the generalization bound with standard empirical Rademacher complexity bound Li et al. (2022a).

**Theorem 3** *For a 1-Lipschitz loss l, with confidence at least $1 - 2\delta$ and for all $f \in \mathcal{F}$, we have*

$$\mathcal{L}^\mu(f) \leq \hat{\mathcal{L}}^\mu(f) + 2\mathcal{R}_{mn}(\mathcal{F}) + 2\mathcal{R}_n(\mathcal{F}) + 3\sqrt{\frac{\ln(2/\delta)}{2mn}} + 3\sqrt{\frac{\ln(2/\delta)}{n}}$$

*where $\mathcal{R}(\mathcal{F})$ standard empirical Rademacher complexity on function class $\mathcal{F}$.*

Now we show that both the number of domains $n$ and the number of images observed from each domain $m$ is negatively correlated to the upper bound of generalization error.

**Proof 2** *Let $P = \{p^{(1)}, \cdots, p^{(n)}\}$ be a set of n domain distribution i.i.d. sampled from $\epsilon$. Define*

$$\Phi(P) = \sup_{f \in \mathcal{F}} \mathcal{L}^\epsilon(f) - \frac{1}{n} \sum_{j=1}^{n} \mathcal{L}_{p^{(j)}}(f) \tag{23}$$

*We construct $P'$ by replacing any $p^{(j)} \in P$ with $p' \sim \mu$, then we have $|\Phi(P) - \Phi(P')| \leq 1/n$. Thus, McDiarmid's inequality tells us that with confidence at least $1 - \frac{1}{2}\delta$*

$$\Phi(P) \leq \mathbb{E}_{P^{(1:n)} \sim \mu}[\Phi(P)] + \sqrt{\frac{\ln(2/\delta)}{2n}} \tag{24}$$

*Following the proof techniques in Theorem 2, we bound the expected value of $\Phi(P)$*

$$\mathbb{E}_{P^{(1:n)} \sim \mu}[\Phi(P)] \tag{25}$$

$$= \mathbb{E}_{P^{(1:n)} \sim \mu}\left[\sup_{f \in \mathcal{F}} \left( \mathbb{E}_{q \sim \mu}[\mathcal{L}_q(f)] - \frac{1}{n} \sum_{j=1}^{n} \mathcal{L}_{p^{(j)}}(f) \right)\right] \tag{26}$$

$$\leq 2\mathbb{E}_{P^{(1:n)} \sim \mu} \mathbb{E}_{(x_j, y_j) \sim p^{(j)}}[\mathcal{R}_n(\mathcal{F})] \tag{27}$$

*McDiarmid's inequality can be used to say with confidence $1 - \frac{1}{2}\delta$ that*

$$2\mathbb{E}_{P^{(1:n)} \sim \mu} \mathbb{E}_{(x_j, y_j) \sim p^{(j)}}[\mathcal{R}_n(\mathcal{F})] \leq 2\mathcal{R}_n(\mathcal{F}) + 2\sqrt{\frac{\ln(2/\delta)}{2n}} \tag{28}$$

*Thus, we have*

$$\Phi(P) = \sup_{f \in \mathcal{F}} \mathcal{L}^\epsilon(f) - \frac{1}{n} \sum_{j=1}^{n} \mathcal{L}_{p^{(j)}}(f) \tag{29}$$

$$\leq \mathbb{E}_{P^{(1:n)} \sim \mu}[\Phi(P)] + \sqrt{\frac{\ln(2/\delta)}{2n}} \tag{30}$$

$$\leq 2\mathcal{R}_n(\mathcal{F}) + 3\sqrt{\frac{\ln(2/\delta)}{2n}} \tag{31}$$

*With Theorem 2, we have with confidence at least $1 - \delta$ that,*

$$\frac{1}{n} \sum_{j=1}^{n} \mathcal{L}_{p^{(j)}}(f) \leq \hat{\mathcal{L}}^\mu(f) + 2\mathcal{R}_{nm}(\mathcal{F}) + 3\sqrt{\frac{\ln(2/\delta)}{2nm}} \tag{32}$$

*Finally, we have*

$$\sup_{f \in \mathcal{F}} \mathcal{L}^\epsilon(f) \leq \frac{1}{n} \sum_{j=1}^{n} \mathcal{L}_{p^{(j)}}(f) + 2\mathcal{R}_n(\mathcal{F}) + 3\sqrt{\frac{\ln(2/\delta)}{2n}} \tag{33}$$

$$\leq 2\mathcal{R}_{nm}(\mathcal{F}) + 3\sqrt{\frac{\ln(2/\delta)}{2nm}} + 2\mathcal{R}_n(\mathcal{F}) + 3\sqrt{\frac{\ln(2/\delta)}{2n}} \tag{34}$$

*which completes the proof.*

Then we prove our Theorem 1.

| Caltech101 | VOC2017 | fairytale | pixel art | wild west |

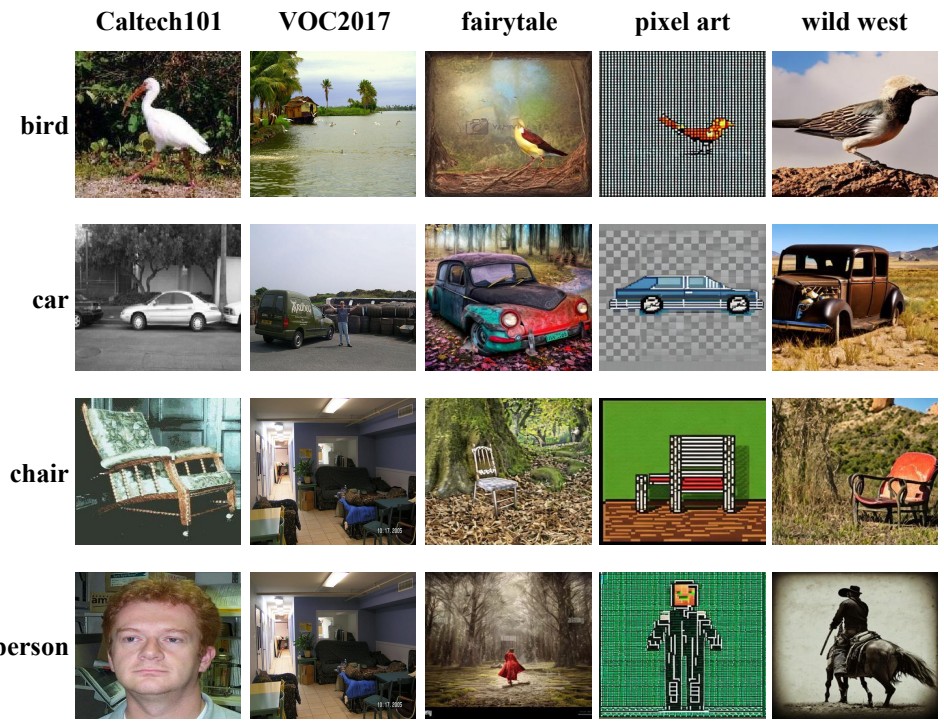

Figure 6: Examples of synthetic images conditioned on novel domain knowledge from LLM. The first two columns (i.e. Caltech101 and VOC2017) are selected from VLCS datasets while the rest three columns are images generated based on the novel domains (i.e. fairytale, etc) provided by LLMs

**Proof 3** *With confidence at least $1 - 2\delta$ and for all $f \in \mathcal{F}$, we have*

$$\mathcal{L}^{\mu}(f) - \hat{\mathcal{L}}^{\mu'}(f) = \mathcal{L}^{\mu}(f) - \mathcal{L}^{\mu'}(f) + \mathcal{L}^{\mu'}(f) - \hat{\mathcal{L}}^{\mu'}(f) \tag{35}$$

*With Theorem 3, we have*

$$\mathcal{L}^{\mu}(f) - \mathcal{L}^{\mu'}(f) + \mathcal{L}^{\mu'}(f) - \hat{\mathcal{L}}^{\mu'}(f) \tag{36}$$

$$\leq 2\mathcal{R}_{mn}(\mathcal{F}) + 2\mathcal{R}_n(\mathcal{F}) + 3\sqrt{\frac{\ln(2/\delta)}{2mn}} + 3\sqrt{\frac{\ln(2/\delta)}{n}} + \mathcal{L}^{\mu}(f) - \mathcal{L}^{\mu'} \tag{37}$$

$$\leq 2\mathcal{R}_{mn}(\mathcal{F}) + 2\mathcal{R}_n(\mathcal{F}) + 3\sqrt{\frac{\ln(2/\delta)}{2mn}} + 3\sqrt{\frac{\ln(2/\delta)}{n}} + \sup_f |\mathcal{L}^{\mu}(f) - \mathcal{L}^{\mu'}| \tag{38}$$

*With the assumption that $D(\mu, \mu') = \sup_f |\mathcal{L}^{\mu}(f) - \mathcal{L}^{\mu'}| \leq \epsilon$, we have*

$$\mathcal{L}^{\mu}(f) - \hat{\mathcal{L}}^{\mu'}(f) \tag{39}$$

$$\leq 2\mathcal{R}_{mn}(\mathcal{F}) + 2\mathcal{R}_n(\mathcal{F}) + 3\sqrt{\frac{\ln(2/\delta)}{2mn}} + 3\sqrt{\frac{\ln(2/\delta)}{n}} + \epsilon \tag{40}$$

*which finishes the proof.*

## B  VISUALIZATION

We provide more examples of synthetic images conditioned on novel domain knowledge from LLM. We present in Figure 6 the synthetic images of VLCS datasets.

## C  PITFALL OF TEXT-TO-IMAGE GENERATION MODELS

Text-to-image generation models are by nature noisy as no strict control can be achieved. We present some pitfalls (commonly reported by the community) that will insert noise and influence the training

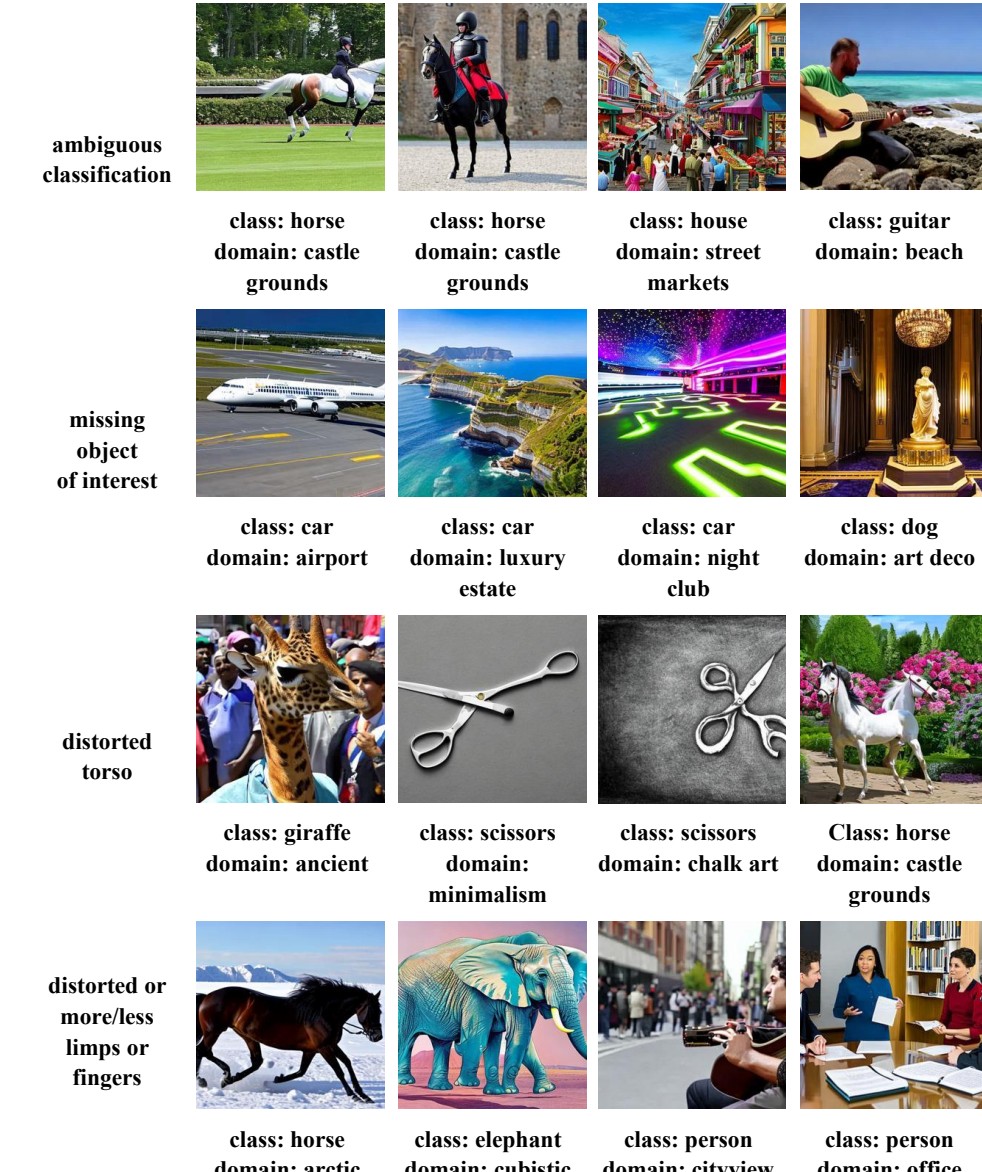

Figure 7: Examples of pitfalls of synthetic images.

of a generalizable model. We show in Figure 7 where each row is a type of problem and below each image is the corresponding class and domain.