# OpenReview forum: "Beyond Finite Data: Towards Data-free Out-of-distribution Generalization via Extrapolation"
_ICLR.cc/2025/Conference — ICLR 2025 Conference Withdrawn Submission_

### Official Review · Reviewer_vThn · 2024-10-23

**Soundness:** 3
**Presentation:** 3
**Contribution:** 1
**Rating:** 3
**Confidence:** 4

**Summary:**

They propose a new approach for domain generalizable image recognition, specifically, leveraging LLM and stable diffusion to expand the knowledge in novel domains. They first propose to generate prompts for novel image domains using LLM and feed the prompts to stable diffusion to generate diverse images. In their experiments, their proposed approach shows superior performance over baseline methods.

**Strengths:**

1. The idea of using LLM and text2image models to augment the image domain is reasonable to boost the performance in generalizable image classification.
2. Their approach shows improvements in experiments.

**Weaknesses:**

1. They claim that their approach is "data-free". But, I disagree with this point since they employ a stable diffusion model to augment data, and the stable diffusion model is trained on a large amount of data. I think their approach shows high performance, especially in the domains in which the stable diffusion is good at generating. Therefore, I think the data to train the stable diffusion includes their target domain. In this sense, I think their approach is still "data-dependent".
2. Their way of generating images from the text2image model is reasonable, yet a bit straightforward and is not very novel, considering the existing work like [1].
3. Compared to the approach like [1], their prompt design relies only on the heuristic knowledge of the dataset. This is because their approach does not feed any training images into their framework. Therefore, if the training images look totally different from images generated from stable diffusion, their approach can fail.


[1] Diversify Your Vision Datasets with Automatic Diffusion-Based Augmentation

**Questions:**

1. It is better to put results other than art or cartoon-like image domains to highlight the robustness of the approach. I think improving the performance on such domains using stable diffusion does not make a much impact due to the reasons I described in the weaknesses 1.

---

### Official Review · Reviewer_Q9t9 · 2024-10-28

**Soundness:** 2
**Presentation:** 2
**Contribution:** 1
**Rating:** 3
**Confidence:** 4

**Summary:**

The paper proposes a method leveraging LLMs to synthesize new domains to aid domain generalization (DG). Experiments are shown to demonstrate benefits.

**Strengths:**

The paper is well written. Ablation studies are good. The idea of using LLMs for DG is somewhat interesting.

**Weaknesses:**

1. Lots of missing baselines to compare against. Please compare and cite the following relevant works:

- P. Teterwak et al, ERM++: An Improved Baseline for Domain Generalization, https://arxiv.org/abs/2304.01973, ICML 2023 Workshop on Spurious Correlations, Invariance and Stability
- C. Liao et al, Descriptor and Word Soups: Overcoming the Parameter Efficiency Accuracy Tradeoff for Out-of-Distribution Few-shot Learning, CVPR 2024
- Vishaal Udandarao et al., SuS-X: Training-Free Name-Only Transfer of Vision-Language Models, ICCV 2023
- L. Dunlap et al, Using Language to Extend to Unseen Domains, ICLR 2023

Namely, ERM++ reports an average accuracy of 81.4% in your experimental setup Table 1 (see Table 1 in ERM++ paper). The Descriptor/Word soup paper has been shown to outperform WaffleCLIP and other GPT descriptor baselines, and is currently SoTA. The SuS-X paper is another missing baseline that uses task information to retrieve data and train models. The LADS paper performs domain extension using language to improve accuracy on target domain using language description as input.

2. Lack of key datasets. The authors have not provided results on DG methods on ImageNet and its variants (Adversarial, Sketch, Rendition, V2), and WILDS distribution shifts. The datasets presented are not challenging for DG and saturated in the literature.

3. It's not clear that such a complex method is warranted to obtain these marginal gains. The method uses large LLMs (GPT4) in addition to text to image generation models (Stable Diffusion 2).

4. The authors need to consider the training data used to train the LLM and text-to-image generation model. Namely, addressing important limitation such as what happens when the data is so different, such as a task on satellite image classification for example, or biology problems, when the LLM and image generation is on Internet-scale datasets. It is well known that on such fine grained specific problems, CLIP-trained models are quite poor.

**Questions:**

See weaknesses.

---

### Official Review · Reviewer_qs4H · 2024-11-01

**Soundness:** 2
**Presentation:** 2
**Contribution:** 2
**Rating:** 3
**Confidence:** 3

**Summary:**

The work pertains to domain generalization/OOD generalization in that it seeks to generate new samples from novel domains using existing generative models.

The proposed pipeline entails:1. input a task description with requests for output format into an LLM. 2. The LLM will then generate prompts that will then be used in an image generation model. 3. The image generation model will generate new images based on these prompts and the images can then be used to train the model on the task.

The work further suggests that this can be done without any previous data. I.e. the task description will be used to prompt the pipeline and no other data than the one generated is used for training the model.

The results of the empirical investigation show that on DomainBed the method performs well and outperforms other augmentation baselines on VLCS and PACS. There are also some additional small ablation studies of e.g. changes to the query strategy and choice of LLM.

**Strengths:**

- The idea is interesting, augmenting datasets with generated examples may be useful in small sample regimes

- The procedure seems to work well for some real-world image datasets

**Weaknesses:**

- The theoretical argument is fairly weak, the assertion that an LLM can approximate the meta distribution over samples is unfounded

- Unclear how many samples are generated and how many samples the model has access to. Does the model using your method have access to more samples than the other baselines?

- Generating more samples may help the model if the samples are of sufficient quality. The method does not guarantee this and seems entirely dependent on the generation performing well.

- Unclear how many times the experiments are run, it is also difficult to compare results as the models have seen different samples

- Experiment on the scaling benefits somewhat unclear, which dataset is being considered?

- Variance analysis on PACS only does not really say much of the variance we can expect in other tasks

**Questions:**

- What happens when the images are from more niche datasets like from the healthcare domain? Do you expect that a diffusion model would generate accurate x-ray images, for example?

- What kind of performance is expected for classes which are underrepresented in the image generator?

- Is there any basis for the claim that you can estimate $\mu$ with an LLM?

---

### Official Review · Reviewer_nE43 · 2024-11-01

**Soundness:** 2
**Presentation:** 3
**Contribution:** 2
**Rating:** 3
**Confidence:** 3

**Summary:**

In this work, the authors propose a novel technique to address challenges in the field of domain generalization. They use an LLM to imagine novel domains over which they generate synthetic data with a diffusion model. They then train models on this synthetic data and find that these models generalize better to new domains in multiple settings. Finally, they even extend to "data free" settings where they train a model from scratch on synthetic data and find that it applies well to real-world data.

While I appreciate the theoretical and empirical results, I am concerned with how realistic these results are and how fair are the author's comparisons. Specifically, I am not convinced that the performance does not come from training models on more data when adding in synthetic domains. I am also unsure how fair it is for large models that may have been trained on the domain's data to then be used in a "data free domain adaptation" setting. Finally, I would love to see some analysis or study of Assumption 1.

**Strengths:**

* The authors create an interesting pipeline for generating synthetic data in novel domains.
* The authors provide some theoretical analysis of the domain generalization problem and provide classical complexity results in the domain generalization setting.
 * The authors provide empirical evidence that their pipeline trains models that generalize better in terms of accuracy, and provide ablation studies of various components of their pipeline. They also analyze components of the pipeline independently, such as the choice of LLM and knowledge extraction approach.
 * I liked your variance analysis experiment.
 * You position your work well in the context of prior literature (such as interpolation failing to effectively increase the number of domains seen)

**Weaknesses:**

* A big weakness of this paper is the key assumption, that $\mu$ and $\mu'$ are close for a language model and the true domain. The authors are able to completely attribute generalization loss to $\epsilon$ and ignore it for the rest of the paper. However, assuming that an LLM does a good job of approximating $\mu$ is a large claim to make. I think it would be good to see some sort of attempt at measuring $\epsilon$, such as even seeing how much this distance changes between two estimated $\mu', \mu''$ generated by two independent LLMs.
 * The authors then say this error can be recovered by increasing $m$, and $n$, but by how much? Would this be a realistic or computationally-feasible amount of data? From an experiment perspective, the baseline methods must be fairly compared with similar values of $m$ and $n$. This is another question I had with the results, but how much more data did your method see than baselines? How would results change if you fixed the same amount of data for baselines, ERM and your method? Otherwise, I do not feel these are fair comparisons.
 * In practice, the LLM and DM are trained on large internet data. It is highly likely that they have seen the image datasets used in this paper (for the DM) or a textual description of these domains. This may provide evidence that the $\epsilon$ is actually quite small, but then raises another question: is it fair to leverage the knowledge of these models for domain generalization given they have seen the generalizing domains?
 * I disagree that data-free generalization can democratize machine learning. Eliminating real-world data collection just results in our representation and learning of certain groups more biased and creates feedback loops where our generative models create biased data that we use to train biased models. Consider, for instance [1], where the authors find that diffusion models generate stereotyped images of various objects, traits, and ethnicities (See Fig. 1 for an illustrative example). In this case, relying on a diffusion model to generalize to a new domain (ie. images from Africa) will not improve performance for users in Africa but instead create harmful stereotypes and potentially not improve model performance.
 * When you increase your domains, do you keep the total number of samples seen constant? Otherwise, it seems difficult to say that it was the domains that led to more performance, rather than just training a model on more data.
 * In your analysis you mention "LLM-generated prompt further extracts knowledge" and later that "the importance of knowledge injected by LLMs that benefits generalization". Can you elaborate on this? Its a pretty vague sentence to me and I would like to understand more clearly what your main thesis is with regards to the importance or contribution of LLMs to domain generalization.

Minor suggestions:
 * You should bold the best scores in your tables, especially Table 1 as it is so large.
 * Line 317, Evaluation is capitalized
 * When you use quotes, make sure you use ``text'' so that latex formats the quotes properly.

[1]: Bianchi, Federico, et al. "Easily accessible text-to-image generation amplifies demographic stereotypes at large scale." Proceedings of the 2023 ACM Conference on Fairness, Accountability, and Transparency. 2023.

**Questions:**

* How does Definition 1 relate to ideas from optimal transport? Why did you define this as your measure of distance as opposed to more traditional OT measures?
 * How much data does each method see in your experiments?
 * How do prompts compare between your LLM generated prompts (your method) and class template/class prompt? I wonder if some of your improved performance comes from better data diversity due to better diversity of prompts when generated by an LLM as opposed to using a template.

---

### Official Review · Reviewer_8uJE · 2024-11-05

**Soundness:** 2
**Presentation:** 2
**Contribution:** 2
**Rating:** 3
**Confidence:** 4

**Summary:**

This paper introduces a method for generating novel domain data to enhance domain generalization in models. The authors first leverage large language models (LLMs) to create relevant novel domains for each class. For each class and domain, they use LLMs to design prompts tailored for text-to-image models, generating images that represent these new domains. Leveraging the synthetic data from these novel domains, the authors demonstrate improved generalization in both single- and multi-domain settings across benchmarks. Notably, even without real data, models trained solely on the generated dataset achieve competitive performance on a few benchmarks.

**Strengths:**

1. The theoretical framework for modeling data distributions that span multiple domains, along with the provided theoretical bounds, could be of value to the community.
2. The approach of leveraging LLMs to generate class-specific domains and using them to design prompts for text-to-image models, ultimately creating domain data, is both useful and, to the best of my knowledge, novel.

**Weaknesses:**

Please see the questions section for more information.

**Questions:**

Despite its strengths, this work has several weaknesses that I encourage the authors to address:
1. **Lack of Technical Details**: The authors omit several crucial details. For instance, it is unclear what specific domains are generated per class, how many such domains are created, and the total number of final data points. Additionally, how is the "novelty" of these domains ensured? The generated domains could potentially overlap with those in the benchmark datasets, thereby limiting the ability to truly measure domain generalization.

2. **Distinction Between Table 2 and Table 3**: As I understand, Table 2 presents results where the generated data is added to a single existing domain for training, with performance measured on all other domains. In contrast, Table 3 reports results after training on synthetic data alone, with performance measured on all other domains. This distinction is somewhat unclear, and I encourage the authors to clarify this in the manuscript.

3. **Unfair Comparison in Tables 1 and 2**: It is unclear if the results across rows are directly comparable, as the proposed method uses significantly more data than other methods. Wouldn’t a fairer comparison involve keeping the dataset size fixed and sampling accordingly from both existing and synthetic data?

4. **Overclaiming Out-of-Distribution (OOD) Capabilities**: The proposed method, at best, generates “novel” domain data relative to existing benchmarks. However, as text-to-image models are trained on specific data distributions, it remains an open question how well they can create data that truly diverges or extrapolates significantly beyond their training distribution. The authors claim at multiple points that “truly novel data” can be created for any task with only task specifications, yet provide no supporting evidence. I encourage the authors to either substantiate this claim or temper their language accordingly.

I am willing to adjust my score if the authors adequately address these concerns.

---

### Note · Authors · 2024-11-14

I have read and agree with the venue's withdrawal policy on behalf of myself and my co-authors.